# Economic intimate partner violence and its co-occurrence with physical and/or sexual intimate partner violence among youth: Mixed-methods findings from Nairobi, Kenya

Anaise Williams[1,2]*, Yurie Aiura[1], Shannon N. Wood[1,2], Mary Thiong'o[5], Grace Wamue–Ngare[3,4], Peter Gichangi[5,6,7], Michele R. Decker[1,2]

**1** Department of Population, Family and Reproductive Health, Johns Hopkins University Bloomberg School of Public Health, Baltimore, United States of America, **2** Johns Hopkins Center for Global Women's Health and Gender Equity, Baltimore, United States of America, **3** Department of Sociology, Gender and Development Studies, Kenyatta University, Nairobi, Kenya, **4** Women's Economic Empowerment Hub, Kenyatta University, Nairobi, Kenya, **5** International Centre for Reproductive Health-Kenya, Nairobi, Kenya, **6** Technical University of Mombasa, Mombasa, Kenya, **7** Department of Public Health and Primary Care, Faculty of Medicine and Health Sciences, Ghent University, Ghent, Belgium

\* awill137@jhu.edu

## Abstract

Partner economic abuse is an understudied form of intimate partner violence (IPV) that is increasingly considered to coincide with physical and/or sexual IPV (i.e., contact IPV), and can hinder help-seeking. This overlap is not well-understood, and measurement for economic IPV remains limited across global settings. This mixed methods study examines 1) measurement performance of an economic IPV scale; 2) prevalence of current economic IPV and contact IPV; 3) demographic and economic agency associations with groupings of IPV experience (mutually exclusive experience, co-experience, and none); and 4) underlying norms that perpetuate economic IPV via qualitative data. Cross-sectional analysis among young, partnered women in Nairobi (n = 591) drew from the 2023 data collection wave of an ongoing youth cohort. Exploratory factor analysis examined measurement performance of the economic IPV scale. Overlap between economic and contact IPV was explored. Multivariable multinomial regression examined associations of sociodemographic and economic agency measures with the groupings of IPV experience. Thematic analysis explored in-depth interview data (n = 15). The economic IPV scale had strong performance. Approximately 35% reported economic IPV, 28% reported contact IPV, and 19% reported both. Economic dependency was associated with increased risk of experiencing economic IPV by itself (RRR: 2.49 95%, CI: 1.15,5.40), and was not similarly associated with contact IPV. Being the primary earner was protective for economic IPV only as compared to no IPV; however, it was a risk factor for combined IPV as compared to economic IPV on its own (RRR: 6.01, 95% CI: 2.01,18.00). Qualitative findings describe a gendered context inclusive of partner financial control. Substantial

**Data availability statement:** Data are currently being finalized for public access through Synapse at the following link: https://www.syn-apse.org/Synapse:syn64693209/wiki/630845.

This work was supported by Bill and Melinda Gates Foundation: PMA/Agile 2.0 Gender/GBV Evidence for Action (INV-046501 to MRD, PG). The funders had no role in study design, data collection and analysis, decision to publish, or preparation of the manuscript.

**Competing interests:** The authors have declared that no competing interests exist.

co-occurrence of economic and contact IPV has implications for survivor financial barriers to help-seeking that cannot be ignored in service provision. Given identified differences in risk between IPV types, economic empowerment programming must apply nuanced attention to unique needs by IPV type.

## Background

Gender-based violence is a multi-faceted phenomenon deeply rooted in gender inequality and gender norms in which women and girls are disproportionally burdened [1]. Intimate partner violence (IPV) is a form of gender-based violence involving acts of physical aggression, sexual coercion/violence, psychological abuse, and controlling behaviors [2]. More than one in four women ages 15 and above globally have experienced IPV at least once [3]. Partner economic abuse has been identified as a form of IPV within the past couple decades yet remains understudied, particularly in low- and middle-income contexts [4,5]. Further, overlap between physical and/or sexual IPV (i.e., contact IPV) and economic IPV has not been widely explored across settings, though mechanisms linking these two forms of violence have been identified [6]. For instance, economic dependency caused by economic IPV can be an obstacle to survivors leaving abusive partners who use contact IPV, increasing risk of re-victimization [7,8]. Conversely, evidence also suggests that contact IPV survivors underperform in workplaces [9], contributing to a negative, escalating cycle of economic dependency on abusive partners that can manifest as economic IPV [7,10–12]. While economic IPV is not limited to low-income couples, household economic instability can lead to conflict, which can lead to increased risk for both economic and contact IPV [5,9,13].

While no universal definition exists, economic IPV often refers to behaviors that control a partner's ability to acquire, use, and maintain economic resources, which threatens economic security, potential for self-sufficiency, and mental health [14]. Multi-item scales may assist in capturing the full range of economic abuse behaviors and providing more accurate prevalences estimates [15]. Adams et al. have categorized economic IPV into three types: economic exploitation, economic control, and employment sabotage [14]. Economic exploitation involves destroying financial resources or credit, including stealing/using money from a partner's bank account, opening credit lines without permission, and refusing to pay bills and rent. Economic control involves preventing access to bank accounts, credit cards, and shared assets, tracking and controlling the use of money, and denying access to food, clothing, and healthcare services. Lastly, employment sabotage involves preventing spouses from obtaining or maintaining employment. Few validated survey measures for economic IPV exist, and the Scale of Economic Abuse (SEA) and subsequent adaptations was the best available at this time of this study's data collection. The 28-item Scale of SEA, published in 2008 from the United States, was one of the first developed and validated economic IPV scales [14]. More recently, the Revised Scale of Economic Abuse (SEA2) reduced the SEA to 14 items under

two subconstructs: economic control and economic exploitation. SEA2 has been validated in the U.S. and Iran [16,17] and is widely used in research on economic IPV [18], however, no known studies have assessed the performance the SEA in sub-Saharan African populations. A recent review found variability in measurement and no context-specific scale available for the region [19].

Sub-Saharan Africa has the highest IPV prevalence globally [20,21], though current findings on economic IPV prevalence across sub-Saharan Africa vary [19]. The 2022 Kenya Demographic and Health Survey (DHS) found that, among ever-partnered women, over 40% experienced economic, emotional, physical, and/or sexual IPV in their lifetime; in the past-year, 18.6% experienced physical and/or sexual IPV, 22.0% experienced emotional IPV, and 5.0% experienced economic violence [22]. The economic IPV variable used in the DHS is a single item measuring whether a partner restricted, exploited, or sabotaged the respondent's ability to acquire economic resources in their lifetime; in Kenya, 10.7% of women responded affirmatively for lifetime experience, although some scholars suggest that this number may be underreported [23]. In Tanzania, 34% of women were found to report past-year economic violence using a 3-item scale [24]. In Nigeria, items from the SEA scale were used to identify a prevalence of within-current relationship economic abuse of 64.2% among married women; however, this study did not do psychometric testing to validate the use of the SEA scale in this setting [25]. Adolescent girls and young women in Sub-Saharan Africa are at heightened risk of partner violence due to their lower social status and economic vulnerability [26].

To our knowledge, no studies have tested the performance of the SEA scale, or a version of the SEA scale, among Kenyan youth. Further, gaps in knowledge exist in sociodemographic differences between young women experiencing economic IPV only and those experiencing economic IPV with contact IPV- critical information for informing IPV response and practice. Against this backdrop, this study aims to 1) report measurement performance of a shortened, revised version of the SEA2 in Nairobi, Kenya among young women; 2) examine prevalence of economic IPV and overlap with contact IPV; 3) compare differences in sociodemographic and economic agency measures between those experiencing no IPV to those experiencing contact IPV only, economic IPV only, and both IPV types, as well as between those experiencing economic IPV only to those experiencing contact IPV only and both IPV types in order to achieve a full picture of differential risk; and 4) qualitatively explore the underlying gender systems and norms that perpetuate and contextualize economic IPV.

## Methods

### Study overview and sample

This study is a convergent parallel mixed-methods analysis, incorporating both cross-sectional quantitative survey data and qualitative in-depth interviews (IDIs) from a longitudinal cohort in Nairobi, Kenya. The study is primarily quantitative, and we use qualitative data that was collected and analyzed simultaneously with the quantitative data to provide context and deeper understanding of the quantitative results. Quantitative data come from a survey administered by International Center for Reproductive Health Kenya in which participants had the option to complete via self-administered (57.3%), interviewer-administered in-person (28.9%), or interviewer-administered over the phone (13.8%).

A youth cohort of both young men and women was initially recruited using respondent-driven sampling, a chain-based recruitment method in which study participants recruit their peers through numbered coupon distribution. For this analysis, only data from the young women's sample is included. At baseline (Round 1), conducted from June to August 2019, 666 unmarried women participants aged 15–24 years residing in Nairobi County were recruited for the study [27]. By Round 4, conducted from June to August 2023, 610 young women who consented to follow-up from previous rounds were retained, now aged 19–29 years. Additionally, a replenishment sample was added to the cohort in Round 4, targeting participants aged 15–19 years (n = 281 young women). As a result, 831 young women completed the Round 4 survey. Among these 831, 596 reported having a sexual or dating partner in the past year. A further 5 participations had missing values for IPV measures. Therefore, the analytic sample for this analysis is 591.

Within Round 4 data collection, a purposive subsample of the cohort was recruited from 01/07/2023 – 15/07/2023, for in-depth interviews (IDIs); sampling was stratified by gender, age, and experiences of violence, in order to obtain a variety of perspectives. IDIs were conducted by trained interviewers using semi-structured interview guides; the women's IDI guide focused on relationships, perception of risk and safety, help-seeking and access to justice, and household roles and decisions. Importantly, the IDIs did not focus directly on economic abuse or gendered economic behavior; however, discussion on gender roles and partner power dynamics provided themes relevant to the economic IPV analysis that contextualize the broader normative environment through which economic IPV operates. IDIs were conducted in English or Swahili and lasted 45–60 minutes on average. This current analysis draws on IDI data from 15 young women participants.

## Measures

**Economic IPV.** Economic IPV was measured via a subset of four items selected from the SEA2 [16,17]. The number of items was determined to reduce participant burden, and specific items selected for Kenyan youth relevance. Among women who reported having a partner in the past year, the four items asked were: During your relationship, how often did your partner do the following: 1) Keep you from having the money you needed to buy food, clothes, or other necessities; 2) Keep you from having a job or going to work; 3) Keep you from taking out a loan or accessing/opening your mPesa account; 4) Force or pressure you to give him/her your money, without paying you back. Response options included never (= 1), hardly ever (= 2), sometimes (= 3), often (= 4), and very often (= 5). Item (3) was adapted to the local Kenya context to include MPesa (mobile money). The economic IPV scale summed up the item responses, with higher scores indicating higher levels of economic IPV. Any economic IPV was operationalized as a binary variable, indicated by an affirmative response to any of the items reported at least once in the relationship.

**Contact IPV.** Contact IPV measures utilized behavioral assessment per best practices inclusive of two survey items [28]. Specifically, the survey asked young women who were partnered in the past-year: "In the past 12 months, has a partner ever pushed you, thrown something at you that could hurt you, punched or slapped you?" and "In the past 12 months, have you had sex with a partner when you did not want to due to threats, pressure, or force?" The binary measure of contact IPV was coded as "1" if the respondent answered "once," "a few times," or "often" to either of the mentioned IPV survey items and "0" if she responded "never" to both survey items.

**Categorical measure of partner abuse.** A categorical variable merging economic IPV and contact IPV was constructed: 0=neither outcome, 1=contact IPV only, 2=economic IPV only, 3=both/overlap IPV types.

**Independent variables.** Demographic covariates included age [15–28], highest level of education completed (below secondary vs. secondary or higher), currently in school (yes/no), currently married or living with a partner as if married (yes/no), lives with own children (yes/no), paid work past year (yes/no), ability to meet basic needs (yes/no), and household wealth (lowest, middle, highest; self-reported). Respondent economic agency measures included report of any economic dependency on respondent's partner for basic needs (yes/no), respondent as primary household earner (yes/no), and full decision-making control over own earnings (yes/no or no earnings).

## Statistical analysis

**Quantitative statistical analysis.** To ensure the four economic IPV items were together measuring a latent construct of economic IPV in this population, exploratory factor analysis was conducted. First, the inter-item correlation matrix of the four economic IPV items was assessed to ensure all correlations between each survey item within the scale were above 0.3, which is the standard threshold for acceptable inter-item correlation [29]. Then, factor analysis with a polychoric correlation matrix due to categorical response items was run. For the items that fell together based on eigenvalue (> 1) and factor loadings (>0.40), a scale was constructed [30]. Analysis of scale reliability focused on inter-item reliability, using both the Cronbach's Alpha Coefficient and the Omega Coefficient. Prevalence of economic IPV by item and overall was reported.

Overlap of any economic IPV with contact IPV was visualized through a Venn diagram. Significance of the correlation between economic IPV and contact IPV was tested using a tetrachoric correlation. Sample characteristics were explored. Bivariate associations of independent variables and economic IPV and contact IPV were explored separately using a bivariate linear regression for the continuous economic IPV scale and the design-based F statistic for the binary contact IPV measure. Bivariate associations of binary measures of economic IPV only, contact IPV only, both IPV types, and neither were explored through the design-based F statistic. Multivariable multinomial regression was run to explore independent variable associations with the categorical variable of no IPV/ contact IPV only/ economic IPV only/ both. Collinearity was assessed using the Variance Inflation Factor (VIF) and independent variables were dropped, incrementally if the VIF exceeded 5. Among all women, two multivariable multinomial regression models were run: [1] reference group of "no IPV" and [2] reference group of "economic IPV only". In running both models, the following comparisons are assessed: contact IPV only vs. neither (model 1), economic IPV only vs. neither (model 1, 2), both IPV vs. neither (model 1), contact IPV only vs. economic IPV only (model 2), both IPV vs. economic IPV only (model 2).

All analyses were conducted using Stata 17.0 (College Station, TX) with statistical analysis set a priori at $p < 0.05$. The analytic sample was restricted to those with complete data for all economic and contact IPV items (5 cases dropped). Less than 3% missingness was present across independent variables within the analytic sample; as such, sample sizes for multivariable analysis float to accommodate. Sampling weights accommodate the RDS study design using RDS-II (Volz-Heckathorn) weights, post-estimation adjustment based on 2014 KDHS population data (age, sex, education levels), and adjustment for loss-to-follow-up. All presented estimates are weighted unless otherwise noted, and statistical testing accounts for clustering among participants recruited by the same recruiter at baseline.

**Qualitative analysis.** All interviews were audio-recorded, translated, and transcribed into English with quality checks by the US and Kenyan teams. The coding and analysis of transcripts were conducted by four research analysts using Atlas.ti software. The team conducted three rounds of paired coding to evaluate agreement between coders. Differences in coding were reviewed and resolved through discussion to ensure the codebook was applied consistently across all team members and IDI transcripts. Changes to the codebook were made to reflect emergent inductive codes, as needed. Codes on economic violence, time use, disagreement, household decision-making processes, and recommendations for improving young men and women were reviewed using an inductive thematic analysis, and matrices were created based on themes and sub-themes.

**Research ethics approval.** This study was approved by the Ethics Review Committee at Kenyatta National Hospital/ University of Nairobi (P310/06/2020) and the Institutional Review Board at Johns Hopkins Bloomberg School of Public Health (IRB 00012952). All procedures aligned with ethical best practices for sensitive topics including specialized training, privacy protections (auditory privacy screener and protocol), and provision of resource referrals [28]. Participants received transport compensation (500 KES or US$3–4 per survey) after participation. All participants provided informed verbal consent. Consent from guardians was not taken for those age 15–17; Kenya national guidance on conducting adolescent sexual/reproductive health research defines youth ages 15 and over as "mature minors", and notes that mature minors may be able to consent for themselves without a waiver of parental permission [31]. Parental permission could prompt undue logistical burdens for participants, and potentially place them in harm's way (e.g., if parents suspected sexual activity or other risk behavior). All participants could skip any questions they did not wish to answer.

**Inclusivity in global research.** Additional information regarding the ethical, cultural, and scientific considerations specific to inclusivity in global research is included in the Supporting Information (S1 Checklist).

## Results

Almost half (48.5%) of the sample was aged 21–24 at the time of survey, with 11% aged 15–20. About 60% had less than a secondary education (Table 2). Only 12% were currently in school. Though the full analytic sample had a dating or romantic partner in the past-year, only one-third were currently married or living with their partner as if married at the time

of survey. Approximately three-quarters had worked for pay in the past year, but 32.5% reported not being able to meet basic needs and 32.3% were dependent on a partner financially. A quarter reported being the primary household earner, but just over half (55.5%) reported full control over their own earnings, as compared to not having full control or not having earnings.

The economic IPV scale ranged from 4-20 with mean 5.4. Exploratory factor analysis of the four economic IPV items suggested that these items associate with each other to describe a single construct (i.e., economic IPV), via a one-factor solution (eigenvalue 2.77) with strong factor loadings (>0.70) and internal reliability (Alpha=0.78, Omega=0.79) (Table 1). Overall, 34.5% reported any economic IPV in their current partnership; item-specific prevalence was: [1] Keep you from having the money you needed to buy food, clothes, or other necessities (25.5%), [2] Keep you from having a job or going to work (18.2%), [3] Keep you from taking out a loan or accessing/opening your mPesa account (8.9%), and [4] Force or pressure you to give him/her your money, without paying you back (14.6%) (Table 1). Slightly over a quarter (28.0%) reported any past-year contact IPV. Economic IPV and contact IPV were significantly correlated (p<0.001) (Fig 1). Among the full sample, 56.5% reported neither form of IPV, 15.4% reported economic IPV only, 9% reported contact IPV only, and 19.1% reported experiencing both contact and economic IPV (Fig 1, Table 3).

In bivariate analysis, the economic IPV scale was associated with higher age (p=0.02), lower education (p=0.03), not being a student (p<0.01), not being able to meet basic needs (p<0.01), lower household wealth (p<0.01), and not having full control over earnings (p=0.04) (Table 2). In bivariate analysis, contact IPV was associated with less education (p<0.01), not being in school (p<0.01), marriage (p<0.01), living with one's own children (p=0.02), not being able to meet basic needs (p=0.05), lower wealth (p<0.01), and reporting economic dependency on a partner (p=0.03) (Table 2).

In bivariate analysis, having experienced economic IPV without contact IPV was associated with not being the primary household earner (p<0.01) (Table 3). Contact IPV without economic IPV was not bivariately associated with any demographic and economic agency measures. Experiencing both economic IPV and contact IPV was associated with less education (p=0.01), not being in school (p=0.03), and lower wealth (p<0.01) (Table 3). Reporting neither IPV outcome associated with higher schooling (p<0.01), being a student (p=0.05), not being married (p<0.01), no children (p<0.01), able to meet basic needs (p<0.01), higher wealth (p<0.01), and not being economically dependent on a partner (p<0.01) (Table 3). In adjusted models, secondary education was associated with reduced risk of contact IPV, both alone (RRR: 0.38; 95% CI: 0.18, 0.82) and with economic IPV (RRR: 0.38; 95% CI: 0.20, 0.72), but was not associated with economic IPV only (Table 4). Married women were more likely to experience contact IPV only or both outcomes, as compared to economic IPV only. Women who were economically dependent on a partner were more likely to experience economic IPV on its own than no IPV (RRR:

**Table 1. Economic IPV item distributions and scale psychometrics among partnered young women (n=591), weighted.**

| | Never | Hardly ever | Some-times | Often | Very often | Any | Factor loadings |
|---|---|---|---|---|---|---|---|
| | row (%) | | | | | | |
| During your relationship, how often did your partner do the following: | | | | | | | |
| Keep you from having the money you needed to buy food, clothes, or other necessities | 74.5 | 4.8 | 15.6 | 3.3 | 1.8 | 25.5 | 0.75 |
| Keep you from having a job or going to work | 81.9 | 5.0 | 8.6 | 2.4 | 2.1 | 18.2 | 0.90 |
| Keep you from taking out a loan or accessing/opening your mPesa account | 91.1 | 3.5 | 3.2 | 1.6 | 0.7 | 8.9 | 0.82 |
| Force or pressure you to give him/her your money, without paying you back | 85.4 | 4.0 | 7.3 | 2.2 | 1.2 | 14.6 | 0.86 |
| **Any economic IPV** | -- | -- | -- | -- | -- | 34.5 | -- |

Eigenvalue: 2.77.

Alpha: 0.778.

Omega: 0.788.

PLOS Global Public Health

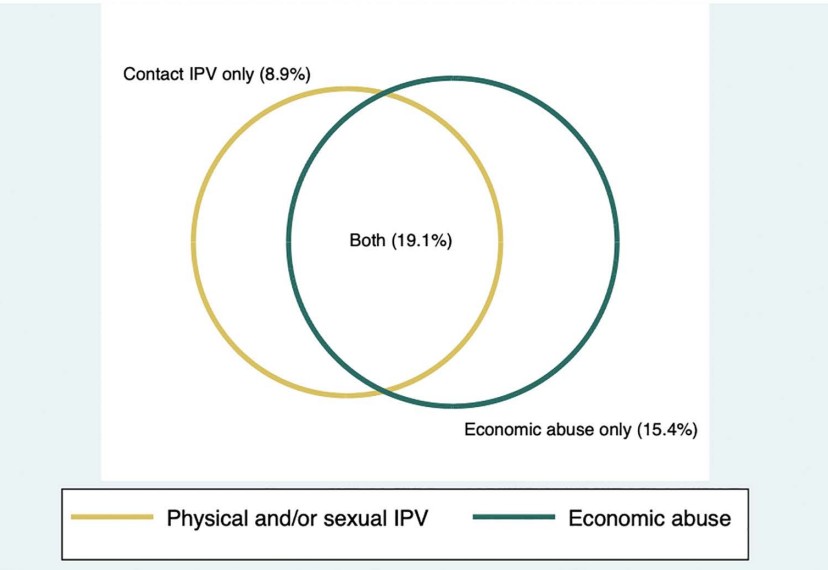

**Fig 1. Any economic IPV during most recent relationship and past 12-months contact IPV with overlap, among partnered young women (n = 591), scaled and weighted.**

2.49; 95% CI: 1.15, 5.40), but economic dependency was not associated with contact IPV, alone or with economic IPV (Table 4). Being a primary earner was protective for only experiencing economic IPV (RRR: 0.32 95% CI: 0.13,0.79) as compared to no IPV, further, primary earners were more likely to experience contact IPV, or both types of IPV, than economic IPV on its own.

## Qualitative results

The quantitative data are informed and contextualized by the broader gendered economic context that young women navigate. Four main themes emerged from the qualitative analysis reflecting and/or supporting quantitative findings: 1) Women's limited economic agency, characterized by gendered labor and decision-making participation, 2) Experiences of economic IPV, 3) economic IPV connections with contact IPV, and 4) women's views on financial independence.

**Women's limited economic agency.** Young women cited salient gendered divisions in financial responsibility and decision-making within romantic partnerships. For instance, informants outlined household divisions of labor characterized by men managing finances and earning money, and women managing other tasks, such as cooking and childcare. In many cases, women are left out of discussion on household finances all together.

> R: [Regarding how we make decisions] *We don't talk about such things, he handles them. I don't even see the money itself; I don't even know how much it is…If I show them it should be paid, he won't argue with me, if I show him the school fees should be paid, he won't argue with me. But when it comes to such decisions it will be used by him and he will make that decision coz the money is not in my pocket.*
>
> -Female IDI participant, ID1, 17 years old, unmarried, contact IPV only

The level of normalization of this division in labor was emphasized by some participants' acceptance of the current system: many women participants expressed comfort with the existing labor division, others showed little interest in questioning it.

**Table 2. Bivariate associations with any economic IPV scale and any contact IPV, among partnered young women (n = 591), weighted.**

| | | Economic IPV | | | Contact IPV | |
|---|---|---|---|---|---|---|
| | Sample col% | Binary indicator row % | Scale Mean [4–20] | p-value+ | Binary indicator row % | p-value++ |
| **Overall** | 100 | 34.5 | 5.4 | -- | 28.0 | |
| **Demographic covariates** | | | | | | |
| Age | | | | **0.016** | | 0.169 |
| 15-20 years | 11.1 | 31.7 | 4.9 | | 17.6 | |
| 21-24 years | 48.5 | 32.1 | 5.2 | | 27.8 | |
| 25-28 | 40.4 | 38.1 | 5.7 | | 31.2 | |
| Highest level of education completed | | | | **0.030** | | **0.001** |
| <Secondary | 60.3 | 40.2 | 5.6 | | 33.9 | |
| >= Secondary | 39.7 | 25.9 | 5.0 | | 19.1 | |
| Currently in school | | | | **0.004** | | **0.003** |
| No | 88.0 | 35.5 | 5.5 | | 30.1 | |
| Yes | 12.0 | 27.2 | 4.8 | | 12.6 | |
| Currently married or living with a partner as if married | | | | 0.313 | | **0.004** |
| No | 67.2 | 30.9 | 5.3 | | 23.2 | |
| Yes | 32.8 | 42.1 | 5.6 | | 38.0 | |
| Has children that she lives with | | | | 0.229 | | **0.015** |
| No | 78.8 | 32.4 | 5.3 | | 25.2 | |
| Yes | 21.2 | 42.2 | 5.7 | | 38.6 | |
| Paid work past-year | | | | 0.269 | | 0.287 |
| No | 27.3 | 37.7 | 5.6 | | 24.4 | |
| Yes | 72.7 | 33.3 | 5.3 | | 29.4 | |
| Ability to meet basic needs | | | | **0.004** | | 0.053 |
| No | 32.5 | 43.4 | 5.9 | | 34.2 | |
| Yes | 67.5 | 30.2 | 5.1 | | 25.1 | |
| Wealth | | | | **0.001** | | **<0.001** |
| Lowest | 31.2 | 43.4 | 6.1 | | 39.3 | |
| Middle | 52.3 | 32.2 | 5.1 | | 26.5 | |
| Highest | 16.5 | 23.5 | 4.8 | | 12.2 | |
| **Respondent economic control and dependency** | | | | | | |
| Economic dependency on partner for basic needs | | | | 0.213 | | **0.027** |
| No | 67.7 | 30.0 | 5.3 | | 24.4 | |
| Yes | 32.3 | 44.0 | 5.6 | | 35.6 | |
| Respondent primary hh earner | | | | 0.327 | | 0.101 |
| No | 74.3 | 35.2 | 5.3 | | 25.6 | |
| Yes | 25.7 | 31.4 | 5.6 | | 34.8 | |
| Full control over own earnings | | | | **0.038** | | 0.795 |
| No or no earnings | 44.5 | 39.4 | 5.6 | | 28.6 | |
| Yes | 55.5 | 30.5 | 5.2 | | 27.5 | |

+ P-value from linear regression continuous economic IPV scale and demographics.

++ P-value of design-based F statistic of binary any contact IPV indicator and demographics.

**Table 3. Bivariate associations with mutually exclusive abuse categories, among partnered young women (n = 591), weighted.**

| | Economic IPV Only (n = 87) | | Contact IPV Only (n = 60) | | Both types of IPV (n = 107) | | None (n = 337) | |
|---|---|---|---|---|---|---|---|---|
| | row % | p-value[+] | row % | p-value[+] | row % | p-value[+] | row % | p-value[+] |
| **Overall** | 15.4 | -- | 9.0 | -- | 19.1 | -- | 56.5 | -- |
| **Demographic covariates** | | | | | | | | |
| Age | | 0.072 | | 0.206 | | 0.244 | | 0.083 |
| 15-20 years | 21.3 | | 7.2 | | 10.5 | | 61.1 | |
| 21-24 years | 11.3 | | 7.0 | | 20.8 | | 60.9 | |
| 25-28 | 18.7 | | 11.8 | | 19.4 | | 50.1 | |
| Highest level of education | | 0.251 | | 0.080 | | **0.012** | | **<0.001** |
| <Secondary | 17.0 | | 10.7 | | 23.2 | | 49.1 | |
| >= Secondary | 12.9 | | 6.2 | | 12.9 | | 68.0 | |
| Currently in school | | 0.505 | | 0.074 | | **0.029** | | **0.053** |
| No | 15.0 | | 9.6 | | 20.5 | | 54.9 | |
| Yes | 18.6 | | 4.0 | | 8.6 | | 68.9 | |
| Currently married or living with a partner as if married | | 0.441 | | 0.054 | | 0.069 | | **<0.001** |
| No | 14.5 | | 6.8 | | 16.4 | | 62.3 | |
| Yes | 17.4 | | 13.3 | | 24.7 | | 44.6 | |
| Has children that she lives with | | 0.477 | | 0.110 | | 0.123 | | **0.005** |
| No | 14.7 | | 7.5 | | 17.7 | | 60.1 | |
| Yes | 17.9 | | 14.3 | | 24.2 | | 43.5 | |
| Paid work | | 0.115 | | 0.220 | | 0.672 | | 0.807 |
| No | 20.0 | | 6.6 | | 17.7 | | 55.7 | |
| Yes | 13.7 | | 9.8 | | 19.6 | | 56.9 | |
| Ability to meet basic needs | | 0.132 | | 0.552 | | 0.060 | | **0.007** |
| No | 19.3 | | 10.0 | | 24.1 | | 46.6 | |
| Yes | 13.5 | | 8.4 | | 16.7 | | 61.4 | |
| Wealth | | 0.873 | | 0.161 | | **0.005** | | **<0.001** |
| Lowest | 16.2 | | 12.1 | | 27.2 | | 44.5 | |
| Middle | 14.2 | | 8.5 | | 18.0 | | 59.3 | |
| Highest | 15.9 | | 4.6 | | 7.6 | | 71.9 | |
| **Respondent economic control** | | | | | | | | |
| Economic dependency on partner | | 0.058 | | 0.199 | | 0.110 | | **0.001** |
| No | 13.0 | | 7.5 | | 16.9 | | 62.6 | |
| Yes | 20.4 | | 12.0 | | 23.7 | | 44.0 | |
| Respondent primary hh earner | | **0.004** | | 0.703 | | 0.132 | | 0.696 |
| No | 18.3 | | 8.8 | | 16.8 | | 56.1 | |
| Yes | 6.8 | | 10.1 | | 24.7 | | 58.5 | |
| Full control over own earnings | | 0.149 | | 0.307 | | 0.315 | | 0.186 |
| No or no earnings | 18.2 | | 7.4 | | 21.2 | | 53.1 | |
| Yes | 13.2 | | 10.1 | | 17.4 | | 59.3 | |

+ P-value of design-based F statistic.

Bolding signifies a p-value below 0.05.

**Table 4. Multivariable multinomial regression associations with mutually exclusive abuse categories, among partnered young women (n = 576), weighted.**

| | Reference group: Neither IPV outcome | | | Reference group: Economic IPV only | |
|---|---|---|---|---|---|
| | Contact IPV only RRR(95% CI) | Economic IPV only RRR (95% CI) | Both RRR (95% CI) | Contact IPV only RRR (95% CI) | Both RRR (95% CI) |
| **Age** | | | | | |
| 15-20 years | Ref | Ref | Ref | Ref | Ref |
| 21-24 years | 0.99 (0.30,3.26) | 0.62 (0.25,1.53) | 2.25 (0.84,6.01) | 1.61 (0.41,6.42) | **3.65\* (1.05,12.72)** |
| 25-28 | 1.46 (0.37,5.73) | 1.35 (0.48,3.69) | 1.81 (0.61,5.39) | 1.09 (0.24,5.02) | 1.34 (0.35,5.13) |
| **Highest level of education** | | | | | |
| <Secondary | Ref | Ref | Ref | Ref | Ref |
| >= Secondary | **0.38\* (0.18,0.82)** | 0.58 (0.29,1.15) | **0.38\*\* (0.20,0.72)** | 0.66 (0.27,1.64) | 0.66 (0.28,1.53) |
| **Currently in school** | | | | | |
| No | Ref | Ref | Ref | Ref | Ref |
| Yes | 0.43 (0.14,1.32) | 1.14 (0.45,2.91) | **0.42 (0.16,1.15)** | 0.38 (0.10,1.42) | 0.37 (0.11,1.24) |
| **Currently married or living with a partner as if married** | | | | | |
| No | Ref | Ref | Ref | Ref | Ref |
| Yes | 2.39 (0.77,7.40) | 0.54 (0.23,1.31) | **2.33 (0.96,5.63)** | **4.38\* (1.19,16.16)** | **4.27\*\* (1.46,12.54)** |
| **Has children that she lives with** | | | | | |
| No | Ref | Ref | Ref | Ref | Ref |
| Yes | 1.70 (0.66,4.35) | 1.69 (0.77,3.69) | 1.12 (0.61,2.07) | 1.00 (0.33,3.02) | 0.66 (0.29,1.53) |
| **Paid work** | | | | | |
| No | Ref | Ref | Ref | Ref | Ref |
| Yes | 1.31 (0.47,3.66) | 1.38 (0.52,3.67) | 1.59 (0.68,3.71) | 0.95 (0.27,3.33) | 1.15 (0.38,3.44) |
| **Ability to meet basic needs** | | | | | |
| No | Ref | Ref | Ref | Ref | Ref |
| Yes | 0.66 (0.29,1.45) | 0.62 (0.32,1.21) | **0.60 (0.33,1.10)** | 1.06 (0.42,2.68) | 0.97 (0.47,1.98) |
| **Wealth** | | | | | |
| Lowest | Ref | Ref | Ref | Ref | Ref |
| Middle | 0.65 (0.29,1.45) | 0.74 (0.37,1.48) | 0.69 (0.36,1.33) | 0.88 (0.34,2.26) | 0.94 (0.43,2.05) |
| Highest | 0.35 (0.10,1.26) | 0.78 (0.28,2.15) | **0.33 (0.10,1.04)** | 0.45 (0.11,1.94) | 0.42 (0.11,1.61) |
| **Economic dependency on partner** | | | | | |
| No | Ref | Ref | Ref | Ref | Ref |
| Yes | 1.02 (0.31,3.35) | **2.49\* (1.15,5.40)** | 1.25 (0.56,2.80) | 0.41 (0.11,1.48) | 0.50 (0.19,1.31) |
| **Respondent primary household earner** | | | | | |
| No | Ref | Ref | Ref | Ref | Ref |
| Yes | 1.07 (0.42,2.76) | **0.32\* (0.13,0.79)** | 1.90 (0.85,4.27) | **3.39 (0.99,11.63)** | **6.01\*\*\* (2.01,18.00)** |

*(Continued)*

**Table 4.** (Continued)

| | Reference group: Neither IPV outcome | | | Reference group: Economic IPV only | |
|---|---|---|---|---|---|
| | Contact IPV only RRR(95% CI) | Economic IPV only RRR (95% CI) | Both RRR (95% CI) | Contact IPV only RRR (95% CI) | Both RRR (95% CI) |
| Full control over own earnings | | | | | |
| No/no earnings | Ref | Ref | Ref | Ref | Ref |
| Yes | 1.21 (0.49,2.95) | 0.76 (0.35,1.66) | 0.62 (0.31,1.26) | 1.58 (0.55,4.54) | 0.81 (0.36,1.84) |

Bold is p<0.10

*p<0.05, **p<0.01, ***p<0.001

RRR=relative risk ratio

*R: I feel comfortable because he is responsible for most of the finances used in the house and doesn't complain about it, so I can't start telling him to start doing the dishes in the house. We agreed that even if I go to hustle, he is still responsible for providing food in the house.*

*-Female IDI participant, ID2, 17 years old, married, IPV data not available*

Roles were ingrained from an early age, shaped by observing parents and older generations.

*R: I think it's okay, I am a girl, I'm supposed to do the laundry, clean the house, the same way my father goes to work, pays the rent and look for food, same as my mum.*

*-Female IDI participant, ID3, 20 years old, married, contact IPV only*

Instances of shared decision-making were less frequent, with variable levels of women's involvement. Disagreements over household decision-making were resolved differently across relationships, although some women participants reported deliberately avoiding direct conflict with their partners. Reflecting evident power dynamics, informants reported often deferring to their partners' choices, particularly around financial decision-making.

**Experiences of economic IPV.** In reflection of the gender norms around women's participation in household financial decision-making and income generation, informants shared experiences of partners seeking to control and, in some cases, directly take ownership of the informant's money. For example, one woman described how a partner forced her to disclose her earnings and savings and neglects her necessities:

*When I save money, he must know. I save money and he takes it and says that even if I am the one who saved it, it's still his money, 'Where did you get that money, is there any work you do? All the money you save is mine?... he leaves little money and it's not enough and yet I have a child. There is no food in the house, he doesn't buy me clothes, I don't have body lotion, hair oil, you see.*

*-Female IDI participant, ID1, 17 years old, unmarried, contact IPV only 1 [This participant's response suggests she has experienced economic IPV though she did not report it in the quantitative survey. While this may be a comment on the validity of the quantitative assessment, it could also be due to the specificity of partner within the question wording. The quantitative measure asks only about the most recent/main relationship, while the qualitative discussion is open to all partnerships (and being unmarried, she could be multi-partnered). It could be that she has not experienced economic IPV in her main partnership, but is describing an experience with a different partner.]*

There were further reports of spouses refusing to support basic needs in the household based on personal preferences, reflecting the norm of male partners having full control over the household's resources.

*If I want something and that something is not good, he always refuses to give me the money to buy it and says that he doesn't like it because it's not good. He makes most of the decisions because he provides the money for most of the things in the house. If I had the money, I would be the one making the decisions.*

*-Female IDI participant, ID2, 17 years old, married, IPV data not available*

As evaluated in the quantitative data, women's descriptions of partner economic control extends to instances where men exert authority over women's choice to participate in income generation. Reflecting gendered financial decision-making in the home broadly, informants discussed trends of male spouses controlling not only decisions on how the household finances are managed, but also the decisions on how women spend their time and allocate their labor (i.e., allocations to household chores versus paid work).

*There was a job I had applied for, but it is far away. Now he was telling me that the job is far away, and I should not go. So, I told him whatever he says, I will go for the job, and he said, 'Let's see between me and you who is the man of the house.'*

*-Female IDI participant, ID2, 17 years old, married, IPV data not available*

**Economic IPV connections with contact IPV.** Discussions on relationships revealed challenges related to couple navigation of financial difficulties. In the most extreme scenarios, these challenges escalated to conflicts where the women were threatened or exposed to physical violence. While physical IPV arising from financial challenges or disagreements was not directly discussed in the interviews, there was one woman who reported physical IPV in parallel with financial pressure from her former partner.

*This guy doesn't go to work, he doesn't do anything. He waits for me to go and hustle and pay the house, feed him, and also pay for my child's school fees, and even if you do all that, he will still hit you. Sometimes he comes home late at night, drunk, he knocks on the door, and since you were sleeping because you are tired, you won't hear him at first. He knocks about twice and when you open the door instead of him coming in first, he slaps you. If you ask him, 'why you are hitting me', he will now beat you more.*

*-Female IDI participant, ID2, 17 years old, married, IPV data not available*

To avoid potential conflict that could escalate to contact IPV, women reported that they tended to follow or agree with their partner's decisions, rather than facing or challenging them.

*M: What happens when you and your husband disagree about anything in your home. When you don't agree about what he says, what happens?*

*R: I just follow what he says.*

*M: Why?.*

*R: I can't say my opinion, if he says something, that's it.*

*-Female IDI participant, ID1, 17 years old, unmarried, contact IPV only*

Sexual favors in exchange for financial support were also discussed among participants. Several young women commented on violence or threats that could happen in transactional relationships.

*You've told him/her you want a certain amount of money and you have already slept with them, or maybe he/she has told you let's do what we are supposed to do then I'll give you the money and he doesn't give you the money, you see that will cause conflict and they will fight.*

*-Female IDI participant, ID4, 19 years old, unmarried, non-partnered*

**Women's views on economic independence.** Within the context of gendered decision-making and male partner financial control described above, participants shared views on the importance of promoting women's financial independence as a key pathway to breaking away from harmful gender norms. Women highlighted that those who have money (typically men) have decision-making power. Many women recognized that securing a job and income could strengthen their decision-making power and, in some cases, provide a means to escape abusive relationships.

*Maybe I should get a job, have my own money, that may help me because if I have my own money, I can make my own decisions. Even if I tell them, you know if you have money that's it. Maybe now you are depending on your husband, there is nothing you can do, you just stay there. If you had your own money, there is nothing they can do to you, you can make all the decisions by yourself*

*-Female IDI participant, ID1, 17 years old, unmarried, contact IPV only*

*R: When you have a job or money so that you can access what you want. Being a woman, it is about money so that you can space some things and figure out what you want in life.*

*-Female IDI participant, ID5, 27 years old, married, econ IPV only*

Others highlighted the broader human rights challenges triggered by gender oppression – suggesting awareness of the gendered systems that limit young women's potential.

*M: And when you look at, generally at the lives of young women, how can they be improved?*

*R: Just given the same opportunities as men.*

*-Female IDI participant, ID6, 25 years old, married, both econ IPV and contact IPV*

## Discussion

In its first validation within sub-Saharan Africa, this study identified strong psychometric performance of an economic IPV scale among young, partnered women in Nairobi, as well as high economic IPV prevalence at 35%, which can be compared to cross-country prevalence ranging from roughly 25–65% [5]. As found in a number of other studies, results highlight a strong correlation and overlap between economic IPV and contact IPV: almost 1 in 5 young women reported experiencing both economic and contact IPV and, among young women experiencing either economic IPV or contact IPV, the majority are experiencing *both* types [5]. Exclusively measuring contact IPV may mask experiences of unique forms of IPV, and specifically economic IPV: about 15% of young women reported experiencing economic IPV without any contact IPV.

While the association of economic IPV and contact IPV has been assessed across contexts, less research has evaluated how sociodemographic risk factors vary by IPV type. This study contributes critical evidence on risks for compounded forms of partner violence. Experiencing either economic IPV and/or contact IPV was associated with IPV risk factors commonly found in the literature, specifically, more limited schooling, married to or living with their partner, children, lower socioeconomic status, and being economically dependent on their partner [32,33]. Risk factors that varied across the two

IPV types included economic dependency, being the prime household earner, higher education, and marriage/cohabitation. While less educated, married/cohabitating women had higher risk of both IPV types than no IPV at all, their risk for economic IPV only as compared to no IPV did not differ from their more educated, non-cohabiting peers. Other research has found the relationship between economic IPV and education to be volatile, with positive and negative associations reported across contexts [5].

In this study, economic dependency was uniquely risky for experiencing economic IPV by itself. Across the IPV literature, economic dependency has been identified as a risk factor for all types of IPV due to becoming trapped in the relationship [12]. This analysis among Nairobi youth finds that while primary household earners are less likely to experience economic IPV by itself, they are much more likely to experience contact IPV, with or without economic IPV, as compared to economic IPV alone. Meaning, primary earners are more likely to experience no IPV than economic IPV on its own, but among those who are experiencing at least one type of IPV, primary earners are more likely to experience contact IPV than economic IPV on its own. This could be related to Male Backlash Theory, in which behaving economically in a way that is not normative, such as, being the primary household earner, has been found to associate with contact IPV across diverse contexts [34]. Taken together, these results suggest that young, partnered women's economic agency and independence may be protective for economic IPV but not similarly associated with contact IPV.

Qualitative evidence contextualizes the highlighted drivers of economic IPV, outlining how young women navigate challenging gender norms that dictate how labor and financial decisions is allocated within the family. Qualitative results also highlight how women navigate household economic decisions, such as through aligning with gender norms around decision-making to avoid conflict. Other studies have found that experiencing economic IPV can lead to women to exercise covert behavior, such as secretly working, which increases risk for contact IPV [5], which may help in part explain the extent of overlap of the two IPV forms in this study.

Strengths of the analysis include the mixed methods approach, in which qualitative results are congruent with quantitative findings. A central limitation of the study is the use of a shortened, revised version of the previously validated SEA2 scale. In using a 4-item subset of the original scale, we are unable to fully compare to other studies employing the SEA or SEA2 scales. There is risk that not all suggested forms of economic IPV are captured. However, results demonstrate that this 4-item shortened scale has strong psychometric performance in this sample. A further limitation related to economic IPV measurement is the referent period, in which respondents were asked about abuse within their current relationship, rather than in the past year as asked about contact IPV. Due to varying relationship lengths, we do not know how recently the economic IPV happened, though the measure was restricted to young women who reported a partner in the past year. Regression analysis is limited by the non-causal nature of the cross-sectional study. The study is further limited by the covariates available in the dataset; relevant relationship and contextual characteristics that were not available include partner demographics, relationship characteristics such as length and level of commitment, and more details on income generating activities.

Recommendations stemming from study findings inform research and policy. For future IPV research, this study confirms strong performance and internal reliability of a revised version of the SEA for use in the Kenyan context among young people. Further, results highlight the need to continue to differentiate forms of IPV. Given distinct risk/protective factors found in this study, future research should avoid grouping contact IPV and economic IPV as a single measure and include both within the portfolio of partner violence. Some evidence suggests that certain forms of microfinance can reduce IPV with even more promising effects of cash transfers [35]. Only a handful of women's economic empowerment program evaluations have explored economic IPV; we need more evidence on effects on economic IPV and pathways for sustainable safety.

Regarding policy, economic IPV has been identified as the form of IPV with the least legislative response across Africa and is often excluded from national action plans [19]. In Kenya, the *National Policy for Prevention and Response to Gender Based Violence* identifies economic IPV as a type of gender-based violence [36], however, the national policy falls

short of providing recommendations specific to economic IPV, thus limiting reach and operationalization. Actionable policies encouraging social norms change around gendered labor and economic control in relation to partner economic abuse is needed. Specifically, in our qualitative results, young women underscored the importance of economic opportunities in their lives and the very real limitations to achieving their economic goals imposed by partners. Policies and programs that encourage expanded women's economic empowerment programming, such as savings accounts, livelihoods trainings, and educational stipends, offer encouraging evidence on both enhancing personal finances and reducing IPV [35], and can support IPV survivors with specific needs stemming from economic IPV. In line with the recent 2025 RESPECT framework, gender transformative social norms programming is needed to change strict gender roles around financial decision-making power within partnerships and foster women's economic opportunity and agency [37].

Partner control over young women's financial resources undermines personal autonomy, limits outside opportunities - reflected in the persistent global gender wage gap [38] - and reinforces economic dependence, contributing to persistent cycles of abuse. Addressing the economic and contact IPV interface is a longstanding priority recognized for its potential for mutually reinforcing gain in violence prevention. Future work must continue to examine the interplay between economic and contact IPV to prevent the cumulative burden of partner violence across the life course.

## Supporting information

**S1 Checklist. Inclusivity in global research.**
(DOCX)

## Author contributions

**Conceptualization:** Shannon N. Wood, Mary Thiong'o, Grace Wamue-Ngare, Peter Gichangi, Michele R. Decker.

**Data curation:** Mary Thiong'o, Peter Gichangi.

**Formal analysis:** Anaise Williams, Yurie Aiura.

**Funding acquisition:** Peter Gichangi, Michele R. Decker.

**Methodology:** Anaise Williams, Shannon N. Wood, Michele R. Decker.

**Project administration:** Mary Thiong'o, Grace Wamue-Ngare, Peter Gichangi, Michele R. Decker.

**Writing – original draft:** Anaise Williams, Yurie Aiura.

**Writing – review & editing:** Shannon N. Wood, Mary Thiong'o, Grace Wamue-Ngare, Peter Gichangi, Michele R. Decker.

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
