## [Decision Letter · Decision Letter 0]

11 Jan 2026

PGPH-D-25-02808

Economic intimate partner violence and its co-occurrence with physical and/or sexual intimate partner violence among youth: Mixed-methods findings from Nairobi, Kenya

Dear Dr. Williams,

Thank you for submitting your manuscript to PLOS Global Public Health. After careful consideration, we feel that it has merit but does not fully meet PLOS Global Public Health’s publication criteria as it currently stands. Therefore, we invite you to submit a revised version of the manuscript that addresses the points raised during the review process.

We look forward to receiving your revised manuscript.

Kind regards,

Emmanuel Olamijuwon, Ph.D

Academic Editor

Journal Requirements:

2. Please provide a detailed online Financial Disclosure statement. This is published with the article. It must therefore be completed in full sentences and contain the exact wording you wish to be published.

a) State the initials, alongside each funding source, of each author to receive each grant. For example: “This work was supported by the National Institutes of Health (####### to AM; ###### to CJ) and the National Science Foundation (###### to AM).”

For more information, please go to our submission guidelines:

https://journals.plos.org/globalpublichealth/s/submission-guidelines#loc-financial-disclosure-statement

3. Please ensure that the funders and grant numbers match between the Financial Disclosure field and the Funding Information tab in your submission form. Note that the funders must be provided in the same order in both places as well.

4. Please update your online Competing Interests statement. If you have no competing interests to declare, please state: “The authors have declared that no competing interests exist.”

6. Please provide separate main figure files in .tif or .eps format only and ensure that all files are under our size limit of 10MB.

Additional Editor Comments (if provided):

Reviewers' comments:

Reviewer's Responses to Questions

**Comments to the Author**

1. Does this manuscript meet PLOS Global Public Health’s publication criteria? Is the manuscript technically sound, and do the data support the conclusions? The manuscript must describe methodologically and ethically rigorous research with conclusions that are appropriately drawn based on the data presented.

Reviewer #1: Yes

Reviewer #2: Partly

2. Has the statistical analysis been performed appropriately and rigorously?

Reviewer #1: Yes

Reviewer #2: I don't know

3. Have the authors made all data underlying the findings in their manuscript fully available (please refer to the Data Availability Statement at the start of the manuscript PDF file)?

Reviewer #1: Yes

Reviewer #2: Yes

4. Is the manuscript presented in an intelligible fashion and written in standard English?

Reviewer #1: Yes

Reviewer #2: Yes

5. Review Comments to the Author

Reviewer #1: 1. Introduction: I would have liked some discussion around:

a. how and why you think economic IPV intersects with sexual and physical IPV

b. Differences around household economic status and economic IPV

c. I think we should also talk about gender as well as gender norms as umbrella structures which influence all forms of IPV and bring in the SES as an intersectional variable.

d. Why do you think SEA is the best fit for assessing economic IPV and the overlap with physical and sexual IPV?

2. Methods:

a. Did you collect this data among young men as well or was it only young women?

b. You say you measure economic IPV through a subset of 4 items, but then say you used a 14-item partner economic abuse scale but in the introduction you say you were going to validate the SEA (as per the introduction) which is a 28 item scale?

i. Economic IPV was measured via a subset of four items selected from the SEA2, a 14-item partner economic abuse scale previously validated in the US and Iran – This sentence is confusing.

c. It would be nice to see what the qualitative interviews covered, what questions were asked and what was analyzed?

3. Results:

a. I wouldn’t show the bivariate associations

b. Your results are structured very nicely!

4. Discussion

a. I would I have likely more discussion about how these results compare to literature – what is comparable what is not. Why you are seeing these associations?

b. It would be nice to also have a conclusion.

Reviewer #2: This manuscript measured economic intimate partner violence (IPV) and its co-occurrence with physical and/or sexual intimate partner violence among young women in Nairobi, Kenya. The manuscript is ambitious in its objectives and focussed on economic IPV is an understudied (compared with other forms of IPV). The study applied a mixed methods approach with a strong quantitative part, noting some disconnect of the qualitative evidence in explaining quantitative results. The comments seek to overcome this limitation and strengthen the manuscript further.

Major comments:

Background

• (115-146) The third paragraph encompasses different themes – moving between methods and evidence. Proposing to improve the flow and structure of the narrative beginning with the definition and measurement of economic IPV; tools and adaptations of the SEA in Africa; the prevalence of different forms of IPV in Africa and Kenya; and then highlighting the vulnerability of young women in this context and evidence gap(s)

Methods:

• (166-175) Please report only the number of female participants (Rounds 1 and 4) since male data was not analysed. Clarify who conducted the interviews, where, how long?

• (201-207) Please clarify the number of items used for assessing “Past-year sexual and/or physical IPV”. It appears as if only one question was used for each physical and sexual IPV respectively, which would not be best practice.

• (242-243) Please explain reasons for two different multivariable multinomial regression models and whether the full sample or only sub-sample of those with any IPV experience were included in the model with ‘economic IPV only’ as the reference group (as ‘no IPV’ estimates are not reported).

• (254-261) “Qualitative analysis” appears to be part of “Statistical analysis”, please change level of this sub-heading. Please explain what qualitative data analysis approach used to analyse qualitative data and provide more details about analysis process after coding.

Results - quantitative:

• Consider starting Results with the description of the sample (currently 283-289).

• Kindly explain the results of the “exploratory factor analysis of the four economic IPV items” in accessible (non-statistical) language (273-274).

Results – qualitative:

• Focus: The qualitative data has a strong focus on gender power relations whilst the survey did not assess gender norms and power but demographic and economic agency measures as covariates. To avoid the disconnect, the narrative part may provide illustrative examples of young women’s limited economic agency (e.g., reworking Theme 1), economic IPV (e.g., reworking Theme 2), and the overlap between contact and economic IPV (currently missing); explore why some women experience economic IPV without any contact IPV (currently missing); and illustrate the identified IPV risk factors (currently missing).

• Good to see quotes used to back up the narrative provided. However, the labels are not unique, making it difficult to assess if a few participants were quoted repeatedly (e.g., Female IDI participant, 17 years old, IPV data not available) or if quotes were taken from across the sample. Consider adding participant ID (1-15). See also below.

• Noting an overlap between Themes 1 and 2 and 3 around financial decision making. Themes may build on each other but should be independent.

• Disclosure of economic IPV: Theme 2 provides examples of economic IPV experienced by young women. However, none of the respondents had reported economic IPV in the survey. For example, a participant (375-379) reported economic IPV (item 1+4) whilst the label suggests that she reported only contact IPV in the survey, undermining the validity of the shortened SEA and results.

• Money equals decision making power: Quotes (411-420) are limited in supporting the claim that financial independence provides women with the means to escape 409 abusive relationships (408-409). Instead, these quotes seem to highlight that power to make decisions rests with people who have money – most often men. However, this does not come out clearly in the narrative.

Discussion

• Mainly summary of key findings. Please discuss, compare with existing evidence.

Minor comments

Abstract:

• Aim: Consider reducing length by reporting the research aim instead of the objectives

• Methods: More detail needed, e.g., data collection

• Results: Consider adding a result for Objective 3

• Recommendations: Repeats Results, could be shorter

Background:

• (106-107) “Partner economic abuse has been identified as a form of IPV within the past decade yet remains understudied” needs a Reference.

Methods:

• (177-178) The specific date (01/07/2023-15/07/2023) may not be needed, consider replacing with “in July 2023”.

• (183-184) “Purposive sampling was used to ensure representation across gender, age, and violence experience.” Duplication (178-179), consider removing.

• (188-189) “a 14-item partner economic abuse scale previously validated in the US and Iran (12, 13)” Duplication (128-129), consider removing.

• (194) “mPesa” please add “(mobile money)” and (196) improve the definition for those unfamiliar with mobile money systems.

• (209) Consider a better term, maybe Overlap of IPV.

• (213-220) Consider separating “demographic variables” from “economic agency measures”.

• (225-226) Please explain and reference the threshold for inter-item correlation to be above 0.3.

• (268) Please report a range of USD as the exchange rate fluctuated a lot during this period.

• (269) Please justify why informed consent from minors without consent from guardians was considered ethical/sufficient.

Results - quantitative:

• Consider better way of reporting the “prevalence across economic IPV” (275-277) starting with any economic IPV followed by “kept from having money you need for necessities (25.5%), having a job (18.2%), and taking out a loan (8.9%), and (4) forced to give money without paying you back (14.6%) (Table 1).”

• Also, consider reporting the Economic IPV Scale Mean here, instead of in Methods (198).

• Consider reporting the overlap of different forms of IPV (299-300) at once after reporting prevalence of either form of IPV (278-279) instead of repeating.

Results – qualitative:

• Some quotes are long and somewhat detached from the narrative. For example, quote (340-345) should follow after “women are left out of discussion on household finances all together” (334); M voice could be removed by adding information to the quote “We don't talk about such things (how money is spend), he handles them…”

• Quote (347-350) responds to “…showed little interest in questioning it” (337). Noting that this quote shows some level of participation in decision making as partners agreed the man is responsible for providing food (even if the respondent works).

• Quote (352-354) would follow naturally after “…shaped by observing parents and older generations” (338).

• Quote (362-367) is not specific to financial decision making (360); it could be shortened to state “I just follow what he says… I can't say my opinion, if he says something, that’s it.”

• Quote (381-382) does not add much, consider removing.

• Other quotes do not provide enough information, raising more questions. For example (399-402), how far is the job away – does the respondent have to relocate or commute? Despite the show of male power, he could also be concerned about their relationship and/or her safety.

• Quotes (426-438) provide weak support for the relationship between education, employment and financial independence.

General:

• Consider defining “contact IPV” clearly once at the beginning of the manuscript to reduce repeating introduction of the term (54, 108, 201).

• Please check for typos and edit for conciseness and language.

6. PLOS authors have the option to publish the peer review history of their article (what does this mean?). If published, this will include your full peer review and any attached files.

**Do you want your identity to be public for this peer review?** For information about this choice, including consent withdrawal, please see our Privacy Policy.

Reviewer #1: **Yes:** Astha Ramaiya

Reviewer #2: No

Figure Resubmissions:

---

## [Decision Letter · Decision Letter 1]

22 Mar 2026

PGPH-D-25-02808R1

Economic intimate partner violence and its co-occurrence with physical and/or sexual intimate partner violence among youth: Mixed-methods findings from Nairobi, Kenya

Dear Dr. Williams,

Thank you for submitting your manuscript to PLOS Global Public Health. After careful consideration, we feel that it has merit but does not fully meet PLOS Global Public Health’s publication criteria as it currently stands. Therefore, we invite you to submit a revised version of the manuscript that addresses the points raised during the review process.

We look forward to receiving your revised manuscript.

Kind regards,

Emmanuel Olamijuwon, Ph.D

Academic Editor

Journal Requirements:

Additional Editor Comments (if provided):

Reviewers' comments:

Reviewer's Responses to Questions

**Comments to the Author**

1. If the authors have adequately addressed your comments raised in a previous round of review and you feel that this manuscript is now acceptable for publication, you may indicate that here to bypass the “Comments to the Author” section, enter your conflict of interest statement in the “Confidential to Editor” section, and submit your "Accept" recommendation.

Reviewer #1: All comments have been addressed

2. Does this manuscript meet PLOS Global Public Health’s publication criteria? Is the manuscript technically sound, and do the data support the conclusions? The manuscript must describe methodologically and ethically rigorous research with conclusions that are appropriately drawn based on the data presented.

Reviewer #1: Yes

3. Has the statistical analysis been performed appropriately and rigorously?

Reviewer #1: Yes

4. Have the authors made all data underlying the findings in their manuscript fully available (please refer to the Data Availability Statement at the start of the manuscript PDF file)?

Reviewer #1: Yes

5. Is the manuscript presented in an intelligible fashion and written in standard English?

Reviewer #1: Yes

6. Review Comments to the Author

Reviewer #1: - Abstract says SEA2 on line 58 but doesn’t provide a full-form. It only becomes apparent in the methods (line 65) that this is an IPV scale. Maybe you can bring this description to line 58.

- All my other comments have been addressed.

7. PLOS authors have the option to publish the peer review history of their article (what does this mean?). If published, this will include your full peer review and any attached files.

**Do you want your identity to be public for this peer review?** For information about this choice, including consent withdrawal, please see our Privacy Policy.

Reviewer #1: **Yes:** Astha Ramaiya

Figure Resubmissions:

---

## [Editor Report · Decision Letter 2]

8 Apr 2026

Economic intimate partner violence and its co-occurrence with physical and/or sexual intimate partner violence among youth: Mixed-methods findings from Nairobi, Kenya

PGPH-D-25-02808R2

Dear Ms Williams,

We are pleased to inform you that your manuscript 'Economic intimate partner violence and its co-occurrence with physical and/or sexual intimate partner violence among youth: Mixed-methods findings from Nairobi, Kenya' has been provisionally accepted for publication in PLOS Global Public Health.

Best regards,

Emmanuel Olamijuwon, Ph.D

Academic Editor